# Prenatal and Early Life Exposure to the Danish Mandatory Vitamin D Fortification Policy Might Prevent Inflammatory Bowel Disease Later in Life: A Societal Experiment

**DOI:** 10.3390/nu13041367

**Published:** 2021-04-19

**Authors:** Katrine S. Duus, Caroline Moos, Peder Frederiksen, Vibeke Andersen, Berit L. Heitmann

**Affiliations:** 1Research Unit for Dietary Studies at The Parker Institute, Bispebjerg and Frederiksberg Hospital, Part of the Copenhagen University Hospital, 2000 Frederiksberg, Denmark; Caroline.moos@rsyd.dk (C.M.); eek@garp.dk (P.F.); Berit.lilienthal.heitmann@regionh.dk (B.L.H.); 2Focused Research Unit for Molecular Diagnostic and Clinical Research, Institute of Regional Health Research, University Hospital of Southern Denmark, 6200 Aabenraa, Denmark; va@rsyd.dk; 3Institute of Molecular Medicine, University of Southern Denmark, 5230 Odense, Denmark; 4The Department of Public Health, Section for General Practice, University of Copenhagen, 1017 Copenhagen, Denmark

**Keywords:** inflammatory bowel disease, Crohn’s disease, ulcerative colitis, vitamin D, fortification, fetal programming, prenatal exposure, ecological study

## Abstract

This register-based national cohort study of 206,900 individuals investigated whether prenatal exposure to small extra doses of vitamin D from fortified margarine prevented inflammatory bowel disease (IBD) later in life; whether the risk of IBD varied according to month or season of birth; and finally, whether there was an interaction between exposure to extra D vitamin and month or season of birth. Fortification of margarine with vitamin D was mandatory in Denmark from the mid-1930s until 1st June 1985, when it was abolished. Two entire birth cohorts, each including two years, were defined: one exposed and one unexposed to the fortification policy for the entire gestation. All individuals were followed for 30 years from the day of birth for an IBD diagnosis in Danish hospital registers. Logistic regression analyses were used to estimate odds ratios (OR) and 95% confidence intervals (CI). Odds for IBD was lower among those exposed to extra D vitamin compared to those unexposed, OR = 0.87 (95% CI: 0.79; 0.95). No association with month or season of birth was found. However, estimates suggested that particularly children born during autumn may have benefitted from the effect of small extra doses of vitamin D. This is, to our knowledge, the first study to explore if prenatal exposure to vitamin D from fortification influenced the risk of IBD. Our results suggest that prenatal exposure to small amounts of extra vitamin D from food fortification may protect against the development of IBD before 30 years of age.

## 1. Introduction

Inflammatory bowel disease (IBD) is a chronic immune-mediated disease that can affects the entire digestive tract [1,2]. The two primary forms of IBD are Crohn’s disease (CD) and ulcerative colitis (UC). Approximately 0.5% of Western populations have IBD, and the incidence is rising worldwide [3]. In northern latitude countries, the prevalence is often higher, and, notably, in Denmark, 1% of the population is affected by IBD [4]. Symptoms of IBD vary, but the most common and distressing symptoms are pain, fatigue, and sleep disturbance. Other symptoms include diarrhoea (sometimes bloody), fever, reduced appetite, and weight loss [5]. In children, growth retardation and late puberty may be prevalent [6]. These severe and invalidating symptoms influence physical and mental health, often compromising social- and work-life, thus affecting quality of life [5].

The aetiology of IBD is multifactorial and still not fully understood, although it is often described to involve interactions between genes, environment, and gut microbiome [7]. One environmental factor widely studied in relation to IBD is vitamin D [8]. Vitamin D, also known as cholecalciferol, is a fat-soluble prohormone steroid that has endocrine, paracrine, and autocrine functions [9]. Most patients with IBD have vitamin D deficiency [10]. Living in southern latitudes also appears to be protective. The hypothesis is that the higher exposure to UV radiation at southern latitudes results in higher vitamin D levels that could be protective [7]. Most recently, has it also been documented that vitamin D has immunoregulatory properties [11,12].

A recent review of meta-analyses showed moderate evidence for the association between vitamin D deficiency and IBD risk [13]. This conclusion was based on results from studies comparing serum vitamin D levels between individuals with IBD and healthy controls [14]. Conversely, few studies have examined the relationship between prediagnostic serum vitamin D levels and IBD in children or adults [15,16]. Additionally, no previous studies, to our knowledge, have investigated whether prenatal or early life exposure to extra vitamin D from fortified food, the overall diet, or from supplements may be related to the risk of IBD later in life.

According to the theory of foetal programming [17], low prenatal vitamin D status could alter the risk of IBD later in life. The critical period for developing the digestive system and organs involved in the immune system is thought to be the first and second trimesters [18,19]. Nevertheless, the maturing of these systems occurs later in gestation and during the period after birth [18,20]. The developing foetus’s vitamin D status relies completely on the mother’s vitamin D status, which depends on dietary intake, supplementation, and exposure to sunlight [21]. In a survey among Danish pregnant women, 56% had insufficient total vitamin D intake, despite 68% using vitamin D supplements [21]. This result highlights the relevance of investigating whether low prenatal vitamin D is associated with disease, such as IBD, in offspring. In Denmark, margarine fortification with 1.25 µg vitamin D/100 g margarine was mandatory until 1st June 1985, when the fortification policy was terminated due to a political decision [19,20,21,22,23]. On average, 13% of the population’s vitamin D intake came from fortified margarine [23]. This implies that children born 1–2 years before 1st June 1985 were exposed to a small extra dose of vitamin D during gestation and early in life compared to children born in subsequent years.

Several studies have investigated the association between month or season of birth and IBD development with inconsistent results [24,25,26,27,28,29,30,31,32,33,34,35,36,37,38,39,40]. Month or season of birth can indicate different environmental exposures during gestation and infancy, e.g., sunshine hours as a proxy for vitamin D exposure, or infections, and/or use of antibiotics. In Denmark, during the winter months, the UV radiation is too low to synthesise vitamin D in the skin [41,42]. This observation may be relevant in understanding why there could be an association between incident IBD and being born in summer and autumn. Children born in summer and autumn in northern latitudes have a ‘darker’ early gestation and possibly a lower exposure to vitamin D in this period.

This study aimed to investigate if exposure to a small extra dose of vitamin D from fortified margarine during gestation and in early life would lower the risk of offspring IBD later in life. We also explored whether individuals born in summer and autumn would have a higher risk of IBD and if the extra vitamin D from fortification prenatally would benefit, in particular, individuals born in summer and autumn.

## 2. Materials and Methods

This study is a societal experiment based on the mandatory Danish margarine fortification policy and its abolishment on 1st June 1985. The study is part of the D-tect project, which has been described in detail earlier [43].

All individuals were selected from two full-year birth cohorts from the Danish Medical Birth Registry immediately before and after the fortification policy’s termination. Exposed individuals were born between 1st June 1983, and 31st May 1985; unexposed individuals were born between 1st September 1986, and 31st August 1988. Individuals were either all exposed or all unexposed to the vitamin D fortification policy for the entire nine months of gestation calculated backwards from the date of birth. Of the exposed individuals, some were also exposed to the vitamin D fortification policy during early life (up to 13 months of age)—for simplicity, we only refer to prenatal exposure.

A 15 month washout period (1st June 1985, to 31st August 1986) was added to account for margarine used to prepare foods that could be stored beyond its shelf life. The 15 months consisted of 4 months of margarine shelf-life and an additional 2 months to use up the fortified margarine from households and 9 months of a normal pregnancy. The follow-up period was 30 years from the day of birth. The design of the study is depicted in Figure 1.

By merging this data with the Danish National Patient Registry, we identified individuals diagnosed with IBD (ICD-8 codes 563.01, 563.02, 563.08, 563.09, 563.19, and 569.04 until the end of 1993 and, hereafter, ICD-10 codes K50 and K51). Only individuals with two records of diagnoses were included in the primary analysis to ensure the cases’ validity.

The following covariates were included to increase the precision of the analyses: sex of the offspring (man, woman), season of birth, month of birth, and type of IBD (CD, UC). Season of birth was categorized as: winter: November, December, January; spring: February, March, April; summer: May, June, July; autumn: August, September, October, based on the fluctuating serum-vitamin D levels during the year [43]. Month of birth was categorized as: January, February, March, April, May, June, July, August, September, October, November, and December.

Type of IBD was based on the diagnosis from the second record: The ICD-codes for CD were ICD-8; 563.01, 563.02, 563.08, and 563.09 until the end of 1993 and, hereafter, ICD-10 code K50; the codes for UC were ICD-8; 563.19, and 569.04 until the end of 1993 and, hereafter, ICD-10 code K51. Individuals registered with both an ICD-code for CD and UC at the second record were categorized as unclassified IBD (U-IBD).

### 2.1. Ethics

According to Danish law, ethical approval is not required for register-based studies when data are used for statistical and research purposes only, in the interest of the public. Permission to access data was granted by The Danish Health Data Authority. The Danish Data Protection Agency provided permission to process data (VD-2019–237).

### 2.2. Patient and Public Involvement Statement

There were no patient or public involvement in designing the study, formulating the research question, performing analyses, interpreting the results, or writing the manuscript.

### 2.3. Statistics

Logistic regression analyses were performed for all modelling. To test the overall association of month and season of birth and the interaction effect, likelihood ratio tests were performed. For the logistic regression analyses, odds ratio (OR), confidence intervals (CI), and *p*-values are given. Data management and descriptive statistics were performed using Stata-13. For the sensitivity analyses, OR was estimated in relation to having at least one, rather than minimum of two, records of IBD. Secondly, crude analysis was performed for the two main diagnoses, CD and UC, to test whether associations differed according to IBD type. Finally, season interaction effect analysis using a season definition based on the calendar (winter: December, January, February; spring: March, April, May; summer: June, July, August; autumn: September, October, November), rather than serum–vitamin D levels, was performed. The statistical analyses were performed using R version 3.3.3 (R Foundation for Statistical Computing, Vienna, Austria, www.R-project.org, accessed on 10 April 2019). All analyses were prespecified in a statistical analysis plan, which was read and approved by all co-authors before analyses.

## 3. Results

### 3.1. Study Population

Of 217,249 individuals, 103,606 had been exposed to the fortification policy prenatally, and 113,643 had not. Among those exposed, 1811 died before the end of follow-up, and 2939 emigrated; among those unexposed, 1888 died, and 3711 emigrated. A total of 206,900 individuals remained for the analyses, whereby 875 among the exposed and 1102 among the unexposed developed IBD (Figure 2).

At the end of the 30-year follow-up, 47.8% among those without IBD had been exposed to extra vitamin D from fortification, compared to 44.3% of those diagnosed with IBD (Table 1). More women were diagnosed with IBD (56.9%) than men. No differences in IBD incidence were seen for season or month of birth.

### 3.2. Prenatal Exposure to Extra Vitamin D from Fortified Margarine

Table 2 shows 13% lower odds for incident IBD among those exposed prenatally to extra vitamin D from fortified margarine compared to those not exposed. Adjustment for sex and season of birth did not change the association (Table 2).

Results were essentially similar whether the outcome was defined based on one or at least two IBD records (data not shown). This was similar when the two types of IBD (CD and UC) were analysed separately. The crude OR was 0.87 (95% CI: 0.76; 1.00, *p* = 0.05) for CD among the exposed, compared to the unexposed at the end of the 30-year follow-up. For UC, the crude OR was 0.87 (95% CI: 0.78; 0.98, *p* = 0.02).

### 3.3. Month of Birth and Season of Birth

No overall association with month of birth was found (*p* = 0.18), and the results were essentially similar before and after adjustment (data not shown). Figure 3 shows the odds of developing IBD, compared to January, with possibly small peaks around May and September and possible dips in February and March.

Analyses showed no association between incident IBD and season of birth (Table 3) in the crude or adjusted models (all *p* ≥ 0.17). The estimates’ direction could indicate a protective effect for spring births, compared to winter, and a higher risk among autumn births, compared to winter. In the sensitivity analysis, where we used the calendar definition of seasons, no overall difference between months were seen (all *p* > 0.25). However, there was a tendency for a slightly higher odds ratio for those born in autumn (Sep–Nov), compared to winter (Dec–Feb) OR = 1.13 (95% CI: 1.00; 1.30) (Table 3).

### 3.4. Interaction between Exposure to Extra Vitamin D from the Fortification Policy and Season of Birth

The overall likelihood ratio test showed no interaction between being exposed to the fortification policy and season of birth (*p* = 0.28). However, among the autumn-born, extra vitamin D from the fortification policy reduced the odds of developing IBD (OR = 0.75 (95% CI: 0.63; 0.89)) (Table 4).

## 4. Discussion 

This study found that individuals exposed prenatally to extra vitamin D from the Danish mandatory fortification policy had 13% lower odds of developing IBD before age 30. This study thereby suggests that even a small extra dose of vitamin D from fortified margarine prenatally seemed protective against incident IBD. Experimental data support the plausibility of this association [7,44,45,46]. Vitamin D deficiency during gestation has been shown to make persistent changes in the immune system, with enlarged spleen and thymus seen in animal models. These changes have previously been associated with autoimmune diseases [46].

Contrary to our findings, a previous Danish study found no association between vitamin D levels at birth and the development of IBD before age 18 years [47]. However, as foetal intestine development occurs during the first part of gestation, neonatal vitamin D levels may not capture risk related to low vitamin D during first and/or second trimesters, and, hence, results between this and our study may not be comparable. In our study, subjects were exposed to extra vitamin D during the entire gestational period and into the first year of life. Furthermore, the individuals in our study were followed in relation to the development of IBD over 30 years. The Danish study followed individuals for 18 years, giving different time under risk periods [47].

For the time of birth analyses, no overall association with season or month of birth was found, somewhat in contrast to other [25,26,27,29,30,31,35,36,37,39,40] but not all studies [24,28,32,33,34,38]. However, we did find that that the fortification policy might have been especially beneficial for individuals born in autumn. These individuals may have had low vitamin D levels during the first part of gestation caused by low maternal skin synthesis of vitamin D due to low UV radiation during wintertime [41].

It is noteworthy that, in a population with an overall insufficient vitamin D intake [23], a small extra dose of vitamin D was associated with a lower odds of incident IBD in the offspring. Our study’s effect was modest, potentially due to the low dose of vitamin D used in the fortified margarine, which supplied 13% (3–29%) of daily vitamin D intake on average [23]. There are no records of vitamin D intake in the years following the termination of the fortification of margarine with vitamin D, but food supply statistics show that margarine use remained stable during the mid-1980s [48]. Moreover, it is unlikely that the population’s food choices changed significantly in the short period from 1983 to 1988 [43]. With this background, it is speculated that, with higher intakes of vitamin D in fortified margarine or other foods, the association may have been stronger.

### 4.1. Strengths and Limitations

A strength of this study is the large study population, which supports a broad generalization of the results. In addition, the design using all subjects born in adjacent birth cohorts makes it likely that any risk factors associated with IBD and/or vitamin D deficiency were equally distributed between the two cohorts, and, therefore, these did not need adjustment. This study’s design is also well suited for exploring new hypotheses, as it can be done relatively easy and at a low cost. An additional strength of this study is the definition of IBD cases. The use of the Danish National Patient Register limits diagnosis to patients with contact to a hospital. An older Danish study assessed the diagnoses of CD and UC (using the ICD-8 coding system) in one single county [49]. They found that 94% of CD and UC cases found in the pathology system were registered in the hospital system. Furthermore, the validity of the hospital system for CD and UC was found to be 97% and 90%, respectively [49], supporting the low chances of false negatives and, thereby, the assumption that all included cases in the study had IBD. Danish registers provide a degree of detail about individuals and the outcome, e.g., type of diagnosis, and ensure that there were relatively few lost-to-follow-up in this design, making misclassification less likely and our analyses more accurate.

A limitation to this study is that we only had information about the exposure at a population level. Unfortunately, we had no access to individual dietary data or measured vitamin D levels, which could have strengthened the results. However, it is reasonable to assume that people in the years following the abolishment would have had an overall lower intake of vitamin D than during fortification [23]. If some margarine products were continually fortified with vitamin D in the years after 1985 and sold in Denmark, the exposure difference to vitamin D between the two cohorts would be less than expected, which would have attenuated rather than inflated the observed odds.

It could be argued that our results are a simple consequence of a secular increase in IBD incidence. To explore this further, we estimated the expected rise in incident IBD cases between the two cohorts, based on Danish IBD incidence rates from 1980 until 2013. The expected rise in incident IBD for the general population was 6.4% during the average period of three years between the two cohorts [4]. As we only assess individuals until age 30 years, we also considered age-specific incidence rates. The highest annual incidence rate (relevant for this study) was 6.2% for CD < 15 years. This gives an expected higher odds of CD (in individuals younger than 15 years) of 19.8% among the unexposed, compared to the exposed. This indicates an expected rise in incident IBD between 6.4% and 19.8% in the period between the two cohorts, suggesting that the secular trend in incidence can partly explain our results. However, these estimations are based on a period that includes the timeframe for our study, indicating that the rise in incidence could also be derived from abolishing the mandatory vitamin D fortification of margarine.

### 4.2. Implication of the Study

To our knowledge, this is the first study to explore the association between prenatal and early life exposure to vitamin D from a dietary source and the risk of IBD later in life. The indication of a protective effect from a small extra amount of vitamin D prenatally supports official recommendations to Danish pregnant women to consume vitamin D supplements during pregnancy [21]. Further studies could investigate whether vitamin D supplements before or during pregnancy give similar benefits.

This finding also adds to the debate of whether mandatory fortification of food items with vitamin D should be resumed [23]. More recent Danish surveys show that vitamin D intake is still very low, compared to recommendations, particularly among women [50]. This indicates that there is still a need for initiatives that will increase dietary vitamin D intake.

## 5. Conclusions

In conclusion, our study suggests that the Danish vitamin D fortification policy, providing a small extra dose of vitamin D during gestation, lowered the odds of developing IBD over the subsequent 30 years. It was indicated that an insufficient intake of vitamin D during gestation might contribute to IBD development. However, the result could be biased, due to the rising incidence of IDB. Future studies ought to test this association using a more robust study design that can determine causality. If causality can be determined, it is also warranted to investigate who will benefit most from vitamin D supplementation/fortification in which situations and the optimal doses.

## Figures and Tables

**Figure 1 nutrients-13-01367-f001:**
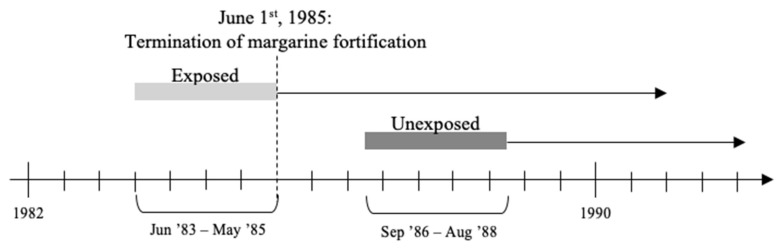
Illustration of study design.

**Figure 2 nutrients-13-01367-f002:**
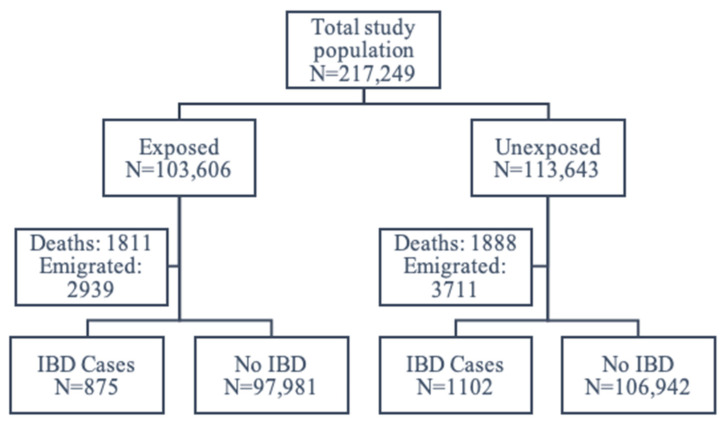
Flowchart of the study population.

**Figure 3 nutrients-13-01367-f003:**
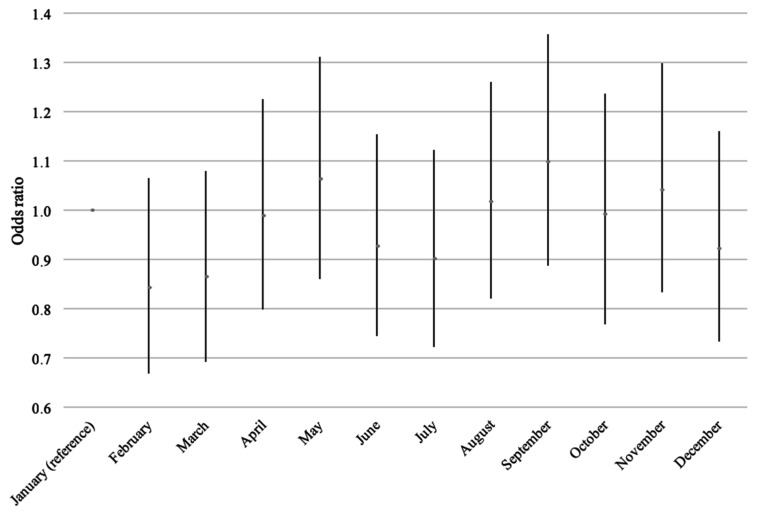
Odds of incident IBD according to the month of birth, adjusted for sex and prenatal exposure to extra vitamin D from the fortification policy.

**Table 1 nutrients-13-01367-t001:** Characteristics of the study population at follow-up (age 30 years), according to the diagnosis of incident IBD.

	No IBD *N* = 204,923 *n* (%)	IBD * *N* = 1977 *n* (%)	*p*-Value χ^2^-Test
Exposed to extra vitamin D ^¤^	97,981 (47.8)	875 (44.3)	0.002
Women	99,898 (48.7)	1124 (56.9)	<0.001
Season of birth			0.17
Month of birth			0.35
Winter	46,299 (22.6)	455 (23.0)	-
November	15,238 (7.4)	158 (8.0)	-
December	15,141 (7.4)	139 (7.0)	-
January	15,920 (7.8)	158 (8.0)	-
Spring	52,360 (25.6)	468 (23.7)	-
February	15,546 (7.6)	130 (6.6)	-
March	18,405 (9.0)	158 (8.0)	-
April	18,409 (9.0)	180 (9.1)	-
Summer	54,644 (26.7)	523 (26.5)	-
May	18,490 (9.0)	195 (9.9)	-
June	18,019 (8.8)	166 (8.4)	-
July	18,135 (8.8)	162 (8.2)	-
Autumn	51,620 (25.2)	531 (26.9)	-
August	18,128 (8.8)	183 (9.3)	-
September	17,129 (8.4)	187 (9.5)	-
October	16,363 (8.0)	161 (8.1)	-
CD cases ^•^	-	816 (41.3)	-
UC cases ^•^	-	1153 (58.3)	-
U-IBD ^•^	-	8 (0.4)	-

* Two records. ^¤^ Vitamin D fortification policy. ^•^ Diagnosis at second record. CD, Crohn’s disease; UC, ulcerative colitis; U-IBD, unidentified IBD (registered with both a CD and UC diagnosis code at the second record).

**Table 2 nutrients-13-01367-t002:** Odds of incident IBD according to prenatal exposure to extra vitamin D from the fortification policy, logistic regression with both crude and adjusted models.

	Crude Model OR (95% CI) *N* = 206,900	*p*-Value	Adjusted Model * OR (95% CI) *N* = 206,900	*p*-Value
Vitamin D fortification policy				
Unexposed	1		1	
Exposed	0.87 (0.79; 0.95)	0.002	0.87 (0.79; 0.95)	0.002

* Adjusted for sex and season of birth.

**Table 3 nutrients-13-01367-t003:** Odds for incident IBD according to season of birth, logistic regression with both crude and adjusted models.

	Crude Model OR (95% CI) *N* = 206,900	*p*-Value	Adjusted Model * OR (95% CI) *N* = 206,900	*p*-Value
Season of birth (vitamin D)		0.17		0.18
Winter (Nov–Jan)	1		1	
Spring (Feb–Apr)	0.91 (0.80; 1.04)	0.15	0.91 (0.80; 1.04)	0.17
Summer (May–Jul)	0.97 (0.86; 1.19)	0.68	0.98 (0.86; 1.11)	0.71
Autumn (Aug–Oct)	1.05 (0.92; 1.19)	0.48	1.05 (0.93; 1.19)	0.46
Season of birth (calendar)		0.25		0.26
Winter (Dec–Feb)	1		1	
Spring (Mar–May)	1.05 (0.93; 1.20)	0.44	1.05 (0.93; 1.20)	0.42
Summer (Jun–Aug)	1.03 (0.90; 1.17)	0.68	1.03 (0.90; 1.17)	0.67
Autumn (Sep–Nov)	1.13 (1.00; 1.29)	0.06	1.13 (1.00; 1.30)	0.06

* Adjusted for sex and exposure to vitamin D fortification policy.

**Table 4 nutrients-13-01367-t004:** Odds for the season interaction effect, logistic regression model with adjustment for sex and exposure to vitamin D fortification policy.

Season of Birth	OR (95% CI)
Winter (Nov–Jan)	0.90 (0.74; 1.08)
Spring (Feb–Apr)	0.90 (0.75; 1.08)
Summer (May–Jul)	0.94 (0.79; 1.12)
Autumn (Aug–Oct)	0.75 (0.63; 0.89)

Likelihood ratio test for overall interaction: *p* = 0.28.

## Data Availability

Data from the D-tecting disease Project cannot be made publicly available for ethical and legal reasons. Public availability may compromise participant privacy, and this would not comply with Danish legislation. Access to the data requires an application submitted to and subsequently approved by the steering committee. Contact Professor Berit L. Heitmann (Berit.Lilienthal.Heitmann@regionh.dk) or the Research Unit for Dietary Studies at The Parker Institute (bfh-eek@regionh.dk).

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
