# Peer review of "Prenatal and Early Life Exposure to the Danish Mandatory Vitamin D Fortification Policy Might Prevent Inflammatory Bowel Disease Later in Life: A Societal Experiment"

_nutrients, 2021, doi:10.3390/nu13041367_

Round 1

Reviewer 1 Report

Review for Nutrients

The investigators used the Danish National Patient Registry to identify individuals with and without IBD, and reconstructed vitamin D exposure, to test the hypothesis that small extra doses of vitamin D from fortified margarine during gestation and in early life reduced the risk of IBD later in life. Exposure reconstruction was based on dates when vitamin D fortification in margarine was policy vs. after. They found that those exposed to extra doses of vitamin D prenatally had a slightly lower risk of IBD. These findings did seem to vary by month.

With 103,606 exposed and 113,643 unexposed individuals, this on is one of the largest study I’m aware of, to evaluate the effects of vitamin D supplementation on IBD, an increasing common disease with debilitating symptoms.  The modeling strategy is simple but rigorous.

This study may be strengthened by the authors addressing the following however.

  1. The large sample size comes at cost of very limited co-variate data. Multiple risk factors have been associated with IBD and/or vitamin D deficiency, but there is no discussion of these factors may influence the findings.
  2. Of more serious concern is the ‘doses of vitamin D exposure’ reconstruction for which time before and after banning margarine fortification—a population level variable. There are no dietary data (self reported) or measured vitamin D in a sample in or outside the cohort, to demonstrate the robustness of the exposure reconstruction. It is also unclear why this analytic strategy is used.
  3. A title that includes ‘vitamin D fortification before and after the ban…’, should replace ‘extra doses’, may be less misleading.

Author Response

  1. The large sample size comes at cost of very limited co-variate data. Multiple risk factors have been associated with IBD and/or vitamin D deficiency, but there is no discussion of these factors may influence the findings.

RESPONSE: Thanks for bringing attention to this. In the discussion we wrote this:

the design using all subjects born in adjacent birth cohorts makes it likely that any confounding variables were equally distributed between the two cohorts and therefore do not need to be adjusted for.

Now, we write:

the design using all subjects born in adjacent birth cohorts makes it likely that any risk factors associated with IBD and/or vitamin D deficiency were equally distributed between the two cohorts, and therefore these did not need adjustment. (page 8, line 250-252)

  1. Of more serious concern is the ‘doses of vitamin D exposure’ reconstruction for which time before and after banning margarine fortification—a population level variable. There are no dietary data (self reported) or measured vitamin D in a sample in or outside the cohort, to demonstrate the robustness of the exposure reconstruction. It is also unclear why this analytic strategy is used.

RESPONSE: We acknowledge this limitation and we have highlighted it in the manuscript.
In the discussion we wrote:

A limitation to this study is that we only had information about the exposure at a population level.

Now, we write:

A limitation to this study is that we only had information about the exposure at a population level. Unfortunately, we had no access to individual dietary data or measured vitamin D levels, which could have strengthened the results. (page 8, line 268-269)

  1. A title that includes ‘vitamin D fortification before and after the ban…’, should replace ‘extra doses’, may be less misleading.

RESPONSE: We understand the concern of the title might be misleading.
We have changed the title from:
‘Prenatal and early life exposure to small extra doses of vitamin D from the Danish mandatory fortification policy might prevent inflammatory bowel disease later in life: A societal experiment’

To:
‘Prenatal and early life exposure to the Danish mandatory vitamin D fortification policy might prevent inflammatory bowel disease later in life: A societal experiment’ (page 1, line 2-3).

Reviewer 2 Report

An interesting population study showing a slight increase in Inflammatory bowel diseases in patients from Denmark not exposed to vitamin D prenatal supplementation. The main limitations of this study are the fact that the results could be biased by the constant increase of IBD incidence and that the results may only be valid for a population with similar characteristics to the danish population. I have some queries:

Materials and methods section, subsection statistical analysis: Please specify the maker and its location of every program used for statistical analysis.

Page 2 line 50 please add: "Vitamin D, also known as cholecalciferol, is a fat-soluble prohormone steroid that has endocrine, paracrine, and autocrine functions. " and add a citation such as: doi: 10.1007/s13668-020-00322-4.

page 1 line 38 this sentence "Inflammatory bowel disease (IBD) is a chronic immune-mediated disease that can affect the entire digestive tract." needs a reference; you may consider this one:doi: 10.1080/03007995.2020.1786681.

Thank You

Author Response

  1. Materials and methods section, subsection statistical analysis: Please specify the maker and its location of every program used for statistical analysis.

RESPONSE: Thanks for bringing attention to this. We have specified this by merging section 2.4 with 2.3., and adding the following on page 4 line 162: (R Foundation for Statistical Computing, Vienna, Austria, www.R-project.org)

  1. Page 2 line 50 please add: "Vitamin D, also known as cholecalciferol, is a fat-soluble prohormone steroid that has endocrine, paracrine, and autocrine functions. " and add a citation such as: doi: 10.1007/s13668-020-00322-4.

RESPONSE: Appreciate the suggestion for elaboration. We have added the suggested sentence (page 2, line 50), with the following reference: P. Borel, D. Caillaud & N. J. Cano. Vitamin D Bioavailability: State of the Art. Critical Reviews in Food Science and Nutrition, 2015, vol. 55 no. 9, pp. 1193-1205, doi: 10.1080/10408398.2012.688897

  1. page 1 line 38 this sentence "Inflammatory bowel disease (IBD) is a chronic immune-mediated disease that can affect the entire digestive tract." needs a reference; you may consider this one:doi: 10.1080/03007995.2020.1786681.

RESPONSE: Thanks for pointing this out. We have added the following references (page 1 line 38):

Kobayashi T, Siegmund B, Le Berre C, et al. Ulcerative colitis. Nature reviews Disease primers 2020;6(1):74. doi: 10.1038/s41572-020-0205-x

Roda G, Chien Ng S, Kotze PG, et al. Crohn's disease. Nature reviews Disease primers 2020;6(1):22. doi: 1038/s41572-020-0156-2

Round 2

Reviewer 2 Report

The authors responded to all queries. The paper may be published